# Hedging our bet on forest permanence for the economic viability of climate targets

Michael G. Windisch [1,2,3] ✉, Florian Humpenöder [1], Leon Merfort [1], Nico Bauer [1], Gunnar Luderer [1,4], Jan Philipp Dietrich [1], Jens Heinke[1], Christoph Müller [1], Gabriel Abrahao[1], Hermann Lotze-Campen [1,2] & Alexander Popp [1,5,6]

Achieving the Paris Agreement's $CO_2$ emission reduction goals heavily relies on enhancing carbon storage and sequestration in forests globally. Yet, the increasing vulnerability of carbon stored in forests to both climate change and human intervention is often neglected in current mitigation strategies. Our study explores modelled interactions between key emission sectors, indicating that accelerated decarbonization could meet climate objectives despite forest carbon losses due to disturbances. However, delaying action on forest carbon loss by just five years consistently doubles the additional mitigation costs and efforts across key sectors, regardless of the assessed forest disturbance rates. Moreover, these myopic responses to forest carbon loss are as stringent, or even more demanding, than immediate responses to twice the forest disturbance rate. Our results underline the urgent need to monitor and safeguard forests for the economic feasibility of the Paris Agreement's climate goals.

The terrestrial carbon sink is critical to mitigate climate change by removing more than 13 billion tonnes of $CO_2$ from the atmosphere yearly, equivalent to a third of all anthropogenic emissions[1,2]. Any conceived climate stabilization target relies on the continuity of this vast carbon removal service. Forests accumulate most of the carbon sequestered this way, constituting two-thirds or 7.8 billion tonnes of $CO_2$ per year[3]. Scenarios limiting warming to well below 2 °C require additional Carbon Dioxide Removal (CDR)[4–7] largely because of hard-to-mitigate or even perpetual emissions sources like concrete manufacturing, aviation, and food production[8–12]. Regrowing forests is the foremost method to provide low-cost and large-scale CDR with present-day technology[13–15].

However, forests are vulnerable to environmental changes and direct human activities such as deforestation and degradation, rendering the permanence of the carbon stored in them uncertain[16–18]. Our exposure to potential forest carbon release is threefold. We rely on 1) the permanence of existing forest carbon stocks, 2) the continued removal of a major fraction of anthropogenic emissions from the atmosphere, and 3) on vastly increasing the global forest carbon stock by large-scale afforestation/reforestation (A/R) to provide us with the much-needed CDR services. Thus, our plan to stay below critical warming thresholds becomes a bet on the world's forests to stay healthy, undisturbed, and productive. Despite that, a growing body of evidence puts the safety of this bet into question.

Recent research reveals that the models informing and simulating today's mitigation pathways might be too optimistic about the future forest carbon storage in three key aspects. 1) Models neglect natural disturbances from windfall, pests, drought, and disease[19] that make up the bulk of current forest damage and show the most pronounced increasing trends in many regions[20–22]. Already modelled disturbances, such as fire, fail to approximate historic trends[23]. 2) Models might overestimate the positive influence of indirect human disturbances of

[1]Potsdam Institute for Climate Impact Research - Member of the Leibniz Association, Potsdam, Germany. [2]Humboldt University of Berlin, Berlin, Germany. [3]Institute for Atmospheric and Climate Science, ETH Zurich, Zurich, Switzerland. [4]Global Energy Systems Analysis, Technische Universität Berlin, Berlin, Germany. [5]Kassel Institute for Sustainability, Kassel, Germany. [6]Faculty of Organic Agricultural Sciences, University of Kassel, Witzenhausen, Germany. ✉e-mail: michael.windisch@env.ethz.ch

warmer temperatures and $CO_2$ fertilization[24]. Global vegetation models agree that ecosystem changes, including shifting ecosystem boundaries, will increase with higher warming levels; however, the specific regions at risk remain uncertain[25]. Findings of Dow et al. in 2022 revealed that warmer spring temperatures do not lead to more carbon being accumulated in temperate forests as projected by models[26]. Further, models project the Amazon rainforest to act as a major carbon sink that may even increase in the future due to higher atmospheric $CO_2$ concentrations[27]. However, more recent studies provide direct evidence of phosphorous limitation potentially cutting biomass carbon growth from $CO_2$ fertilization in half[28,29]. 3) Direct human disturbance like deforestation and degradation are likely underestimated in scenario-building models that currently assume perfect efficiency in replacing forest land with agriculture. A study in 2022 by Pendrill et al. found that only 45% to 65% of deforested land for agriculture was used within a few years after clearance due to mismanagement, land speculation, and uncontrolled fire clearings[30]. Integrated Assessment Models (IAMs) tasked with simulating mitigation pathways may have overestimated the remaining carbon budgets by neglecting these major disturbances. These include natural disturbances (windfall, pests, diseases, etc.), their modulation by indirect human disturbances, such as a warming climate and elevated $CO_2$ levels, and the underestimated influence of direct human disturbances from land expansion and harvest. This potential overestimation of the future forest carbon sink in IAMs suggests the need for precautionary adjustments to modeled mitigation targets.

In this work, we investigate how neglecting major forest disturbances affects mitigation pathways and assess the consequences of postponed action to understand the importance of more accurate forest carbon projections for climate policy. We introduce a range of stylized disturbance rates to the IAM REMIND-MAgPIE (REgional Model of Investment and Development-Model of Agricultural Production and its Impact on the Environment, Supplementary Fig. 1)[31,32]. These rates are designed to represent both direct and indirect human disturbances currently missing in projections of IAMs in a stylized manner. All additional disturbance rates introduced lie within the range of observed rates (see Methods & Supplementary Fig. 7). Moreover, we emphasize findings that remain consistent across the entire range of disturbance rates.

We assess mitigation policy scenarios reaching climate targets despite substantial carbon losses associated with the introduction of stylized disturbance rates to both natural and planted forests. In a set of 54 scenarios, we explore mitigation pathway adjustments along five dimensions: (i) the rate of annual forest disturbance ($n = 4$ & control), (ii) different long-term climate targets ($n = 2$), (iii) varying socio-economic settings ($n = 3$), (iv) multiple climate model projections ($n = 3$), focusing the assessment on (v) the different policy regulations ($n = 5$) (Supplementary Fig. 1). The climate variables and potential carbon densities are derived from established data of General Circulation Models, a Dynamic Global Vegetation Model, and a Simple Climate Model (Supplementary Fig. 1). Informed by these natural boundaries, we explore the importance of knowing more about forest carbon dynamics for societal decisions and the implications of delayed, myopic action.

## Results

### Mitigation response to forest carbon loss
In the following, we contrast the foresighted policy scenario taking immediate action with the myopic response, both aiming at the 1.5 °C-consistent budget under the additional annual disturbance rate of four per thousand trees per year in the middle-of-the-road Shared Socio-economic Pathway 2 (SSP2) scenario (Fig. 1a, b, Supplementary Fig. 7). Despite the associated forest carbon loss (FCL), the land sector remains a net carbon sink until 2055, which roughly coincides with the reaching of net negative $CO_2$ emissions (Supplementary Fig. 4). The

chosen stylized disturbance rate (0.4% per year (yr)) results in the most conservative difference between the foresighted and myopic responses across key mitigation indicators, including the $CO_2$ price, permitted emissions, and land-based mitigation area. The only exception is the land-based mitigation area response, where the 0.2% per yr disturbance rate shows a smaller difference (+103%, Supplementary Fig. 2).

Reaching the 1.5 °C target despite an additional annual disturbance rate of four in a thousand trees is achieved by measures that further lower emissions over all $CO_2$-emitting sectors (Figs. 2 & 3), expanding renewable energy supply (Fig. 3), ramping up CDR deployment (Figs. 2 & 4), and boosting the price on carbon (Figs. 2 & 5). The adjustment to the FCL can be less abrupt and sweeping if the response is foresighted and preemptive (SSP2-1.5 °C-FCL-Foresight). As the time window for mitigation narrows, the regulation needs to become more severe. Thus, a myopic policy response to the same forest disturbance rate (SSP2-1.5 °C-FCL-Myopic) comes at a much greater cost in terms of $CO_2$ prices required to achieve the carbon budget (Fig. 2).

We find the additional mitigation cost and land-based mitigation effort (Fig. 2b, c, Supplementary Fig. 2) more than double responding to the same disturbance rate when comparing the foresighted response to the myopic, five-year-delayed response − a pattern consistent across all assessed disturbance levels (Supplementary Fig. 2). Moreover, the myopic response to FCL imposes the same or even more severe emission reduction requirements, land-based mitigation needs, and overall mitigation costs compared to a foresighted approach dealing with twice the disturbance rate (Supplementary Fig. 2). This finding also remains consistent across all assessed disturbance rates (Supplementary Fig. 2).

### Implications for the energy sector
In the SSP2-1.5 °C scenario without FCL, renewable energy supply is expanded by 122 ExaJoule (EJ) per yr between 2030 and 2050 to limit cumulated emissions to 500 Gigatonnes $CO_2$ (Gt$CO_2$) (Fig. 3a). Consequently, emissions from the energy, industry, transport, and buildings sector are reduced by 69% (Fig. 3c). To achieve a further 40% reduction in emission by the end of the century, an even higher renewable energy share is required (Fig. 3a/c).

Both the foresighted and myopic FCL scenarios project a higher renewables and bioenergy share in the first half of the century (Fig. 3b). Pre-emptive action minimizes the adjustments needed later in the century. In the SSP2-1.5 °C-FCL-Foresight scenario, fossil-fuelled energy sources are further reduced by −5.8% (−17 EJ per yr) by 2030 compared to the scenario without FCL, and yearly emissions decrease by an extra -1.5 Gt$CO_2$ per yr, mainly from the energy supply and industry sectors (Fig. 3d).

In the myopic policy response scenario (SSP2-1.5 °C-FCL-Myopic), the additional renewables capacity in 2050 is tripled, increasing from +2.2 EJ per yr to +7.3 EJ per yr, and fossil-fuelled energy sources are phased out 1.6 times faster (-37.5 EJ per yr) than in the foresighted adjustments (−23.3 EJ per yr) (Fig. 3b). As a result, the emission rate in the myopic scenario is 734Mt$CO_2$ per yr lower than in the foresighted scenario. Two-thirds (492Mt$CO_2$ per yr) of these further emission reductions come from the industry and transport sectors. Additionally, emission reductions in the buildings sector double between the foresighted and myopic scenario (Fig. 3d).

In 2050, permitted emissions under the myopic scenario with the 0.4% annual disturbance rate (4299 Mt$CO_2$ per yr) are even lower−and thus more stringent−than those in foresighted scenario facing double the disturbance rate (0.8% per year, 4365 Mt$CO_2$ per yr; Fig. 2, Supplementary Fig. 3). Across all assessed disturbance rates, permitted emissions for myopic responses remain more stringent than foresighted responses to double the forest disturbance rate (Supplementary Fig. 2).

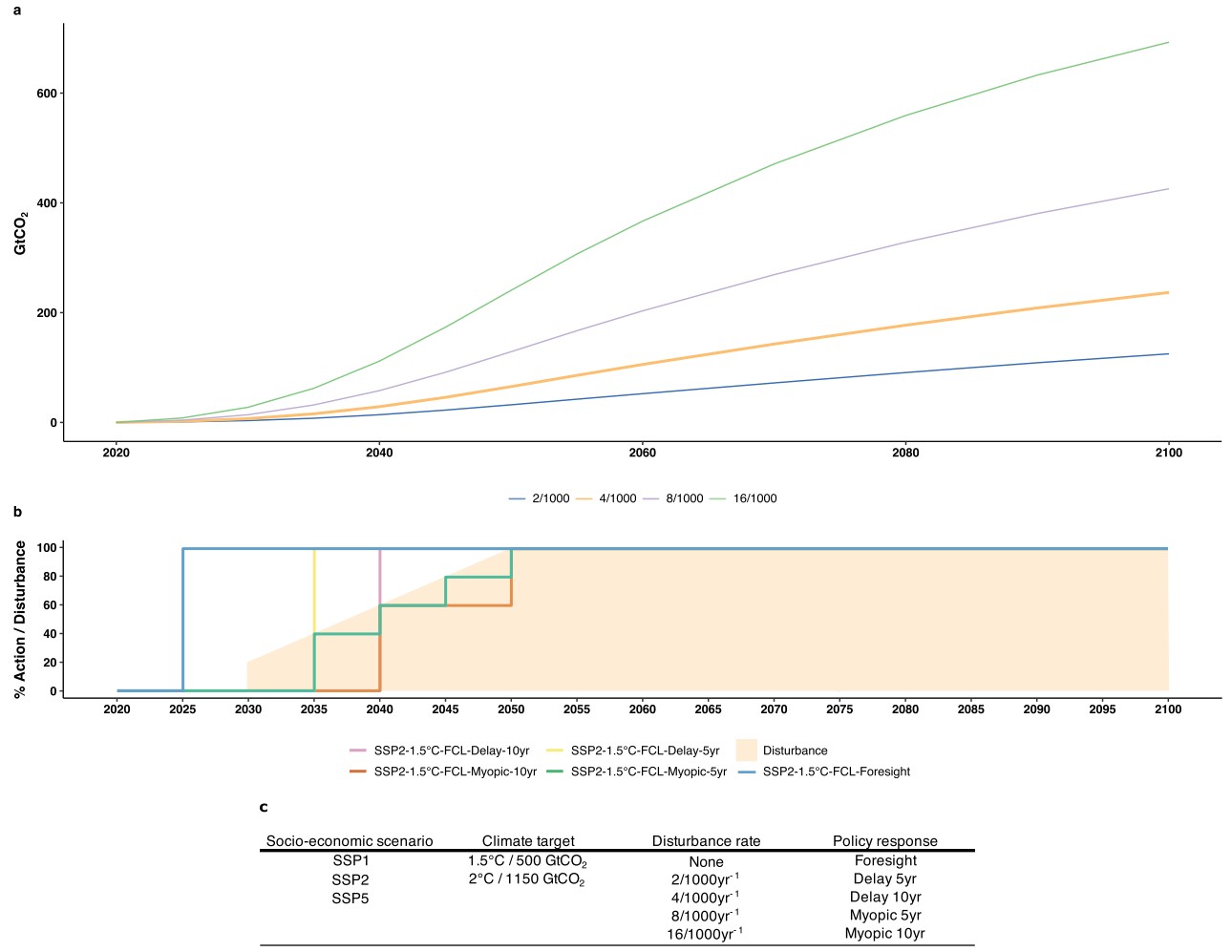

**Fig. 1 | Cumulative forest carbon loss from increased disturbance rates and policy response scenarios. a** Cumulative global net $CO_2$ emission in billion tonnes (emission from additional disturbance minus regrowth) caused by four added, stylized disturbance rates leading to Forest Carbon Loss (FCL): 2/1000, 4/1000, 8/1000, and 16/1000 trees per year (blue, orange, purple, green). Emissions shown are the difference to the projected land carbon sink and subsequently take up part of the carbon budget of the respective scenario. The land carbon sink is projected by LPJmL (Lund-Potsdam-Jena managed Land) & MAgPIE (Model of Agricultural Production and its Impact on the Environment) for human-induced changes, the stylized disturbance rates (shown here) and by MAGICC (Model for the Assessment of Greenhouse Gas Induced Climate) for natural processes (Supplementary Figs. 1 and 4). The 4/1000 disturbance rate focused on in the main text is highlighted in bold. **b** Consists of two parts. The first shows the fraction of the full disturbance rate over time (filled area). All additional disturbance rates are introduced in 2030 and linearly ramp up to their full extent reached in 2050. Superimposed are the timings of the five different policy response scenarios. The percent policy action represents the magnitude of forest carbon loss which the policy action responds to. Any coloured disturbance above a scenario action line indicates a lack of preparedness to handle the corresponding level of forest carbon loss. Blue shows the perfect foresight scenario responding immediately. Yellow/pink policy scenarios respond five/ten years after the initial disturbance with perfect foresight. Green/orange scenarios respond with myopic foresight but with a five/ten-year delay. **c** Explored socio-economic ($n = 3$), climate target ($n = 2$), disturbance rate ($n = 4$ & control), and policy response settings ($n = 5$).

## Negative emissions

Energy-side mitigation measures are accompanied by negative emissions to reach the 1.5 °C target. Technical CDR capacity (BioEnergy- and Direct Air Carbon Capture and Storage (BECCS /DACCS)) in the SSP2-1.5 °C scenario without FCL doubles every 2.6 years from 2022[33] to 2030, adding on average 40 Mt$CO_2$ per yr of additional capacity per year (Fig. 4a).

The SSP2-1.5 °C-FCL-Foresight scenario has a 40% higher yearly growth rate in the first half of the century, with +132 Gt$CO_2$ more cumulative negative emissions by 2100 (Supplementary Fig. 5).

Delayed implementation of negative emission infrastructure in the SSP2-1.5 °C-FCL-Myopic scenario leads to a steeper growth rate in the second half of the century, almost doubling the rate of the foresighted scenario from +100Mt$CO_2$ per yr to +198Mt$CO_2$ per yr (Fig. 4a). The myopic scenario also relies on more DACCS reaching a capacity of 8.6Mt$CO_2$ per yr compared to 3.7Mt$CO_2$ per yr in FCL-Foresight and 1.0Mt$CO_2$ per yr in the SSP2-1.5 °C scenario without FCL. As a result, the cumulative negative emissions in 2100 are +242 Gt$CO_2$ higher in SSP2-1.5 °C-FCL-Myopic compared to the SSP2-1.5 °C scenario, resulting in almost twice the additional cumulative carbon removal than the foresighted policy in response to the same forest disturbance rate (Supplementary Fig. 5).

## Land-use demand

Using land-based methods to remove carbon dioxide from the atmosphere, such as BECCS and A/R, requires vast areas of land. In the SSP2-1.5 °C scenario, 224 million hectares (Mha) of land is reserved for mitigation until 2030 (Fig. 4b). However, this expansion is accompanied by a simultaneous reduction of 247 Mha of crop- and pastureland despite of a growing population creating the need for more efficient agricultural practices. By the time cumulated emissions peak in 2050, land-use for CDR efforts and reduction in crop- and pastureland will almost double. Under the foresighted FCL response, an additional 69Mha of land is allocated for CDR by 2050.

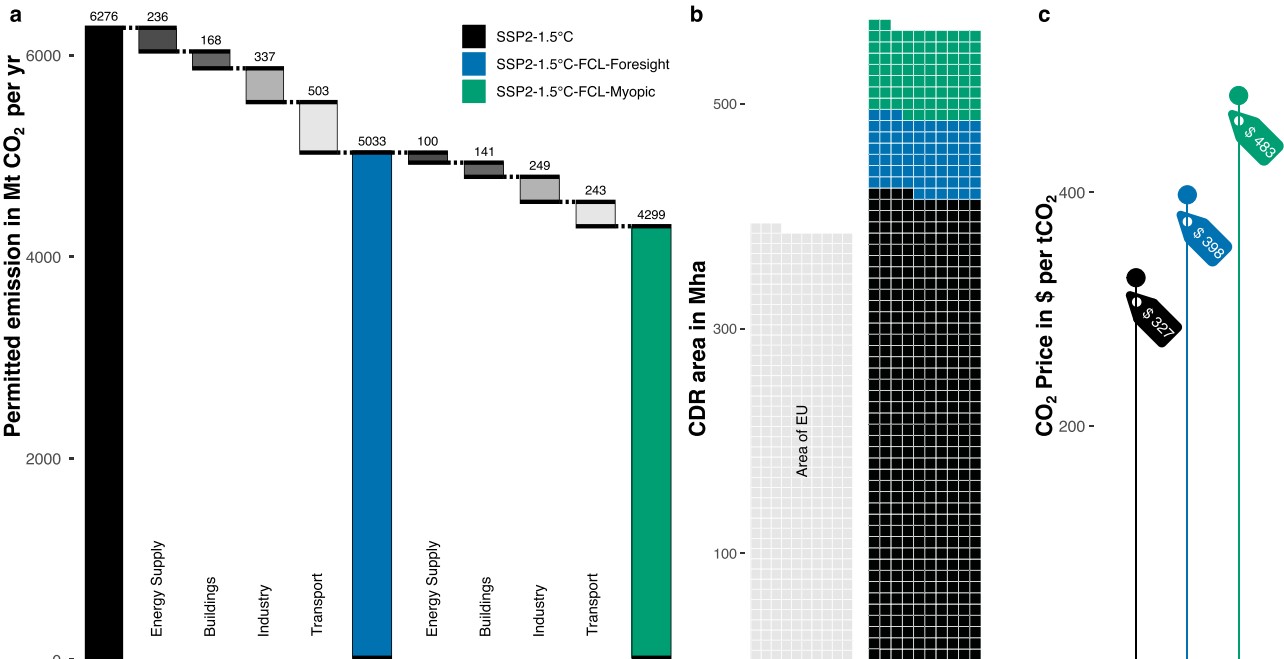

**Fig. 2 | Policy responses to forest disturbance overview in 2050. a** Total gross permitted emission (million tonnes) in 2050 of energy use (coloured by scenario) and emission reductions implemented in response to forest disturbance between three scenarios by sector (greyscale), (**b**) additional area (million hectares) occupied by Bioenergy with Carbon Capture and Storage (BECCS) and Afforestation/Reforestation (A/R) by 2050, and (**c**) the required $CO_2$ price (US dollars per tonne $CO_2$) driving regulations in 2050. Scenarios depicted are the Shared Socio-economic Pathway 2 1.5 °C scenario (SSP2-1.5 °C) without taking forest disturbances into account (black), the scenario mounting a foresighted preemptive response to Forest Carbon Loss (FCL) (blue), and the scenario in which action against the same FCL is taken myopically five years after the initial disturbance (green). For a sense of scale, the total land area of the European Union is added to b).

However, the myopic scenario uses +149Mha more land for mitigation, effectively doubling the additional land-use of CDR increase compared to the foresighted response (Figs. 2 & 4b). By 2050, the required land area for mitigation in the five-year-delayed myopic response to a 0.4% per year disturbance rate reaches 563 Mha, which is even higher than the 542 Mha projected in the foresighted scenario responding to twice the disturbance rate (0.8% per year, Supplementary Figs. 2 & 3). At any assessed disturbance rate, the required land areas for mitigation of the myopic responses are the same or more extensive than the foresighted response to twice their disturbance (Supplementary Fig. 2).

### Economic cost and drivers

Carbon pricing is a central tool incentivizing the growth of renewables and CDR in the model. However, such climate mitigation policies may also slow economic growth measured by the Gross Domestic Product (GDP). The SSP2-1.5 °C scenario requires a carbon price of 115 $ per $tCO_2$ in 2030, rising to 327 $ per $tCO_2$ in 2050 (Fig. 5). The peak GDP loss due to the climate policy is projected to be −2.1% by mid-century compared to a world without strengthened climate policies (Supplementary Fig. 6).

Taking immediate foresighted action on FCL results in a 22% increase in carbon price in 2030 and 2050 and a peak GDP loss of −2.4%. Myopic policy adjustments lead to a 48% increase in the carbon price, with a peak GDP loss of −2.7%. Thus, the additional increase in carbon price and GDP cost incurred by the FCL response more than doubles between the foresighted and myopic scenarios facing the same forest disturbance rate (Figs. 2, 5, & Supplementary Fig. 6). Responding myopically rather than immediately to the FCL consistently requires twice the effort, regardless of the disturbance rate. Additionally, the carbon price of the five-year-delayed myopic response to the 0.4% per year disturbance rate (483$ per $tCO_2$, Fig. 2, Fig. 5) is comparable to the foresighted response to twice the disturbance rate (484$ per $tCO_2$ at 0.8% per year, Supplementary Figs. 2 & 3). The carbon price of the myopic responses are close to equal or higher than the foresighted response to twice their disturbance rate at any assessed disturbance rate (Supplementary Fig. 2).

### Discussion

This study applies a range of highly stylized disturbance regimes, that are within currently observed boundaries found in literature, to the global multi-sectoral model REMIND-MAgPIE. Our results suggest that immediate, ambitious mitigation action is the most effective way to prepare for so-far underexplored consequences of human activity like an increase in forest disturbances. Achieving mitigation goals despite FCL required cuts in projected emissions from all sources, as well as large-scale CDR deployment. Implementing a higher carbon price in the modelled scenarios was crucial to achieving the adjustments to the reference mitigation pathway. Delayed and myopic decarbonization efforts exacerbated the problem, resulting in a more than doubled $CO_2$ price increase (+71$ per $tCO_2$ to +156$ per $tCO_2$), relative GDP loss (−0.25% to −0.63%), and added mitigation area (+69 Mha to +149 Mha). Moreover, the myopic response to FCL imposed the same or even more severe emission reductions, land-based mitigation needs, and overall mitigation costs than the foresighted approach facing twice the disturbance rate. The consistent doubling of key mitigation efforts between the foresighted and myopic responses, as well as the comparability of a myopic response to a foresighted response at twice the FCL, held consistently across all assessed disturbance rates.

Discussed in this study are aspects and shortcomings of global models that estimate current and future carbon emissions and removals from forests informing and simulating mitigation pathways. The carbon flux estimates of these global models are distinctly different from the country-level Nationally Determined Contributions (NDCs) and their tracking by the National Greenhouse Gas Inventories

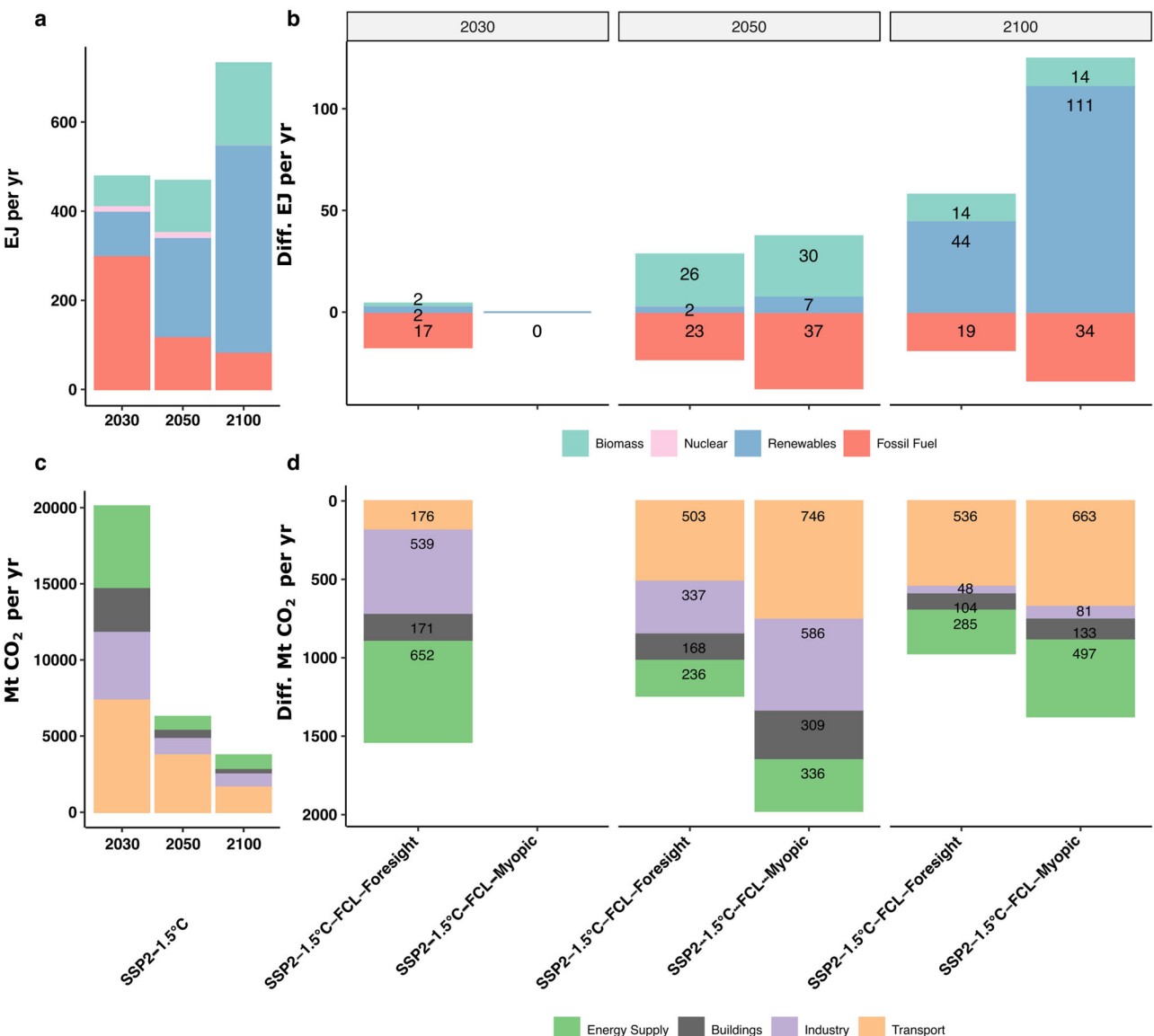

**Fig. 3 | Global energy supply and emissions by sector. a** Annual primary energy supply (Exajoule (EJ)) in the Shared Socioeconomic Pathway 2 1.5 °C scenario (SSP2-1.5 °C) and (**b**) Difference in annual primary energy supply of the foresighted and myopic response to Forest Carbon Loss (FCL) compared to the SSP2-1.5° scenario. Positive/negative values in (**b**) indicate more/less energy supplied compared to the SSP2-1.5 °C scenario shown in (**a**) without FCL. Supply is split into biomass (turquoise), nuclear (pink), renewable energy from hydro-, solar-, wind-, and geothermal sources (blue) and fossil fuel driven energy from coal, oil, and gas (red). **c** Depicts sectorial emission of the SSP2-1.5 °C scenario without forest disturbances differentiating between emissions from supplying energy (green), buildings (grey), industry (purple), and transport (orange). **d** Shows the difference in sectorial emissions between the foresighted and myopic FCL response scenarios and the SSP2-1.5 °C scenario. The negative values in (**d**) indicate reduced sectorial emissions compared to the SSP2-1.5 °C scenario shown in (**c**) without FCL. All results are shown for the years 2030, 2050, and 2100. Please note the differences in the y-axis scales between (**a–d**).

(NGHGIs) as their definition of anthropogenic vs natural forest carbon fluxes differs[34–36]. Global mitigation pathway simulations highlighted in the IPCC reports are produced by IAMs, including the REMIND-MAgPIE model employed in this study. Most commonly, these mitigation pathways of IAMs only consider emissions and removals induced by direct anthropogenic effects such as deforestation, wood harvest, and the regrowth that follows harvest or land abandonment. In contrast, the NGHGIs tasked to track the NDCs consider all carbon fluxes in "managed forests", which serves as a proxy to distinguish between anthropogenic and natural carbon emissions and removals. Thus, the highly stylized disturbance regimes used in this study provide an initial assessment of the role forest disturbances play within the global multi-sectoral mitigation modeling of IAMs. A more detailed understanding of both direct and indirect human influences on these disturbances is essential for incorporating more realistic representations into upcoming multi-sectoral modeling efforts. Additionally, examining different rates of forest disturbances caused by individual drivers, such as pests or windfall—for example, assessing a strong pest versus a strong windthrow response to climate change—could provide more targeted insights to inform management strategies for mitigating specific types of disturbances.

Despite assuming disturbances, A/R can still aid mitigation efforts if their regrowth outpaces emissions from disturbed forests. However, without a conservative (large) buffer fraction, establishing additional forests increases the exposure to future $CO_2$ emissions from disturbances. In addition, a large buffer likely lowers the appeal of A/R establishment as it would increase the price per absorbed $CO_2$. Further, the buffer could exacerbate issues linked to poor A/R decisions

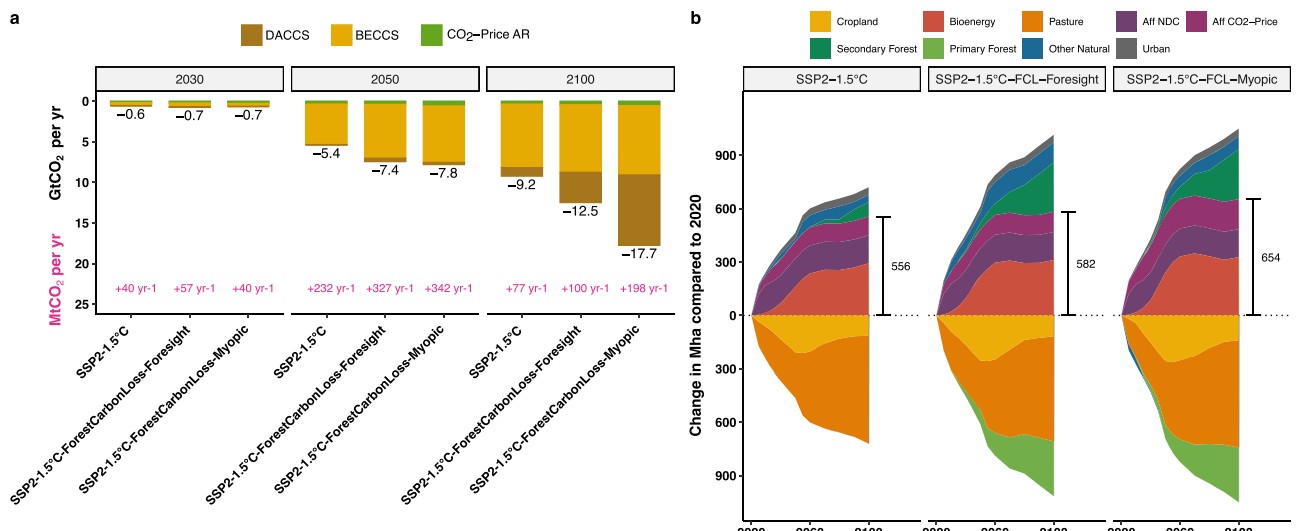

**Fig. 4 | Global negative emissions and land-use change. a** Annual negative emissions in billion tonnes by Direct Air Carbon Capture and Storage (DACCS) (brown), BioEnergy Carbon Capture and Storage (BECCS) (yellow), and A/R (green) in 2030, 2050, and 2100. The Shared Socioeconomic Pathway 2 1.5 °C scenario (SSP2-1.5 °C) is shown next to the two forest carbon loss response scenarios. Yearly growth rates in technical Carbon Dioxide Removal (CDR) capabilities (BECCS and DACCS) in million tonnes are added in pink. Numbers depict the average growth rate necessary between shown points in time. In the case of 2030, technical CDR growth rates are shown for the values reported by the International Energy Agency (IEA) in 2022 (45 MtCO$_2$ per yr installed CCUS capacity[33]) and the model output of 2030. **b** Area difference of main land-use types to their respective 2020 extent in million hectares. The two disturbance response scenarios are shown to the right of the SSP2-1.5 °C scenario without forest disturbances. Afforestation/Reforestation (A/R) actions are split into efforts planned under the Nationally Determined Contributions (Aff NDC) and solely price-driven forest establishment (Aff CO$_2$-Price). The total mitigation land area (both A/R and bioenergy) in 2100 is highlighted in black.

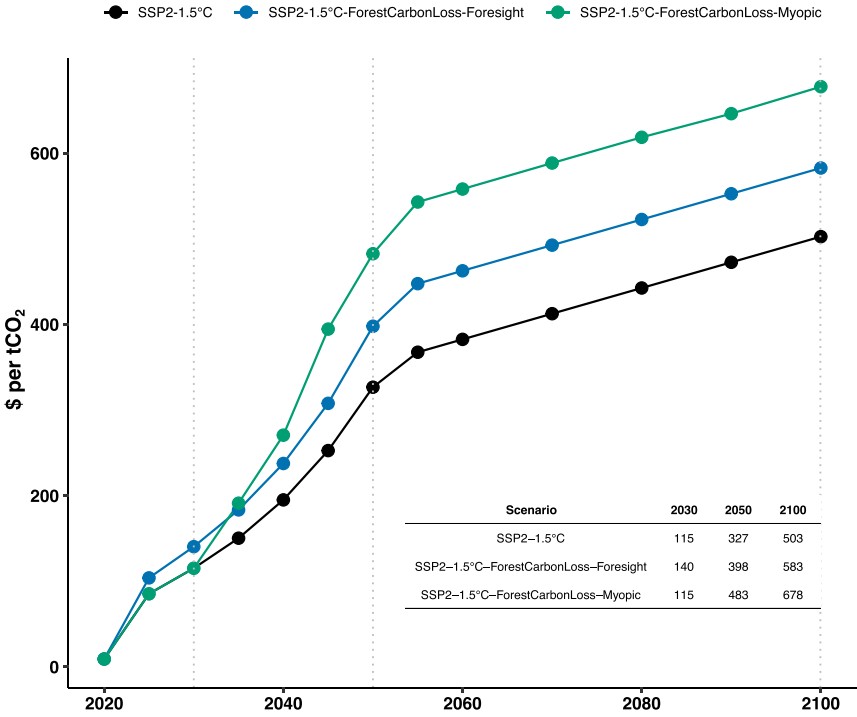

**Fig. 5 | Global CO$_2$-price development.** CO$_2$-price evolution in US dollar ($) per tonne CO$_2$ of the foresighted and myopic disturbance response scenarios (blue, green) and the reference 1.5 °C compatible scenario (black). Prices for 2030, 2050, and 2100 are highlighted in the inserted table.

hiking prices by competing for agricultural land and biodiversity loss of plantations. However, positive feedback could also become more extensive as native forests can bolster species richness[37]. To effectively use forests in mitigation efforts, reliable estimates of future disturbance rates are essential.

The A/R establishment in scenarios presented here is partially exempt from the response policies' delayed and myopic behaviour. As the revenue for A/R is formed over a 50-year planning horizon (see materials and methods), the A/R decision partially reaches beyond the period of fixed policies of the delayed and myopic scenarios. However,

future A/R revenue is discounted by the predetermined, SSP-specific interest rate trajectories (Supplementary Table 1). Thus, short-term revenue affected by the delayed and myopic policies is weighted more in the A/R decision.

Future forest disturbance rates, their spatial distribution, and evolution are still highly uncertain. Some sources, like fire, are better quantified than others, such as windfall and pests[19]. Disturbances from fire are already included in the model chain used here. However, the implemented fire dynamics only lower the average carbon storage and, thus, only indirectly induce emissions by decreasing the potential storage.

Our study aimed to explore the space of societal responses to various disturbance rates resulting from different, unaccounted-for sources of disturbances. However, we did not assess spatial heterogeneities, so we cannot draw any regional conclusions from our results. Additionally, we did not examine differences in the timing of disturbances. In all scenarios, $CO_2$ emissions from forests were introduced before the cumulative emission peak was reached, allowing for mitigation efforts to adjust and achieve their targets in almost all cases. Only scenarios that aimed to limit warming to 1.5 °C with an annual disturbance rate of 1.6% and ten-year myopic actions fail to meet their goal. Introducing forest disturbances closer to the peak would likely result in more scenarios being inconsistent with their initial target. The scenarios explored here anticipate sustained negative emissions after the peak is reached. Thus, any $CO_2$ released from forests late in the century would likely induce less response as the negative emissions form a buffer to the peak of $CO_2$ in the atmosphere.

As signs of decreasing forest resilience become more and more evident, it is crucial to include major forest disturbance processes in global models that estimate the future carbon budget. If these processes are not taken into account, unanticipated emissions from increasing rates of forest disturbances could consume a considerable portion of the remaining emission allowance, potentially jeopardizing mitigation efforts.

## Methods

### Model description

REMIND-MAgPIE combines the energy-economy model REMIND (regional model of investments and development) with the land-use allocation model MAgPIE (model of agricultural production and its impact on the environment). The dynamic global vegetation model LPJmL (Lund-Potsdam-Jena managed land) provides biophysical constraints of the land-use system, which is driven by bias-adjusted climate projections from ISIMIP, based on the CMIP6 earth system model MRI-ESM2, UKESM, and, IPSL-CM6A for SSP119 and SSP126 as provided by the ISIMIP project[38].

The REMIND and the MAgPIE models compute economic equilibrium by combining optimization methods and iterative fixed-point methods. For a given climate target, such as a carbon budget that constrains global cumulative emissions, the coupled model framework derives the optimal strategy given the socioeconomic, technological, political and geobiophysical constraints. This optimal strategy is a portfolio of mitigation measures that consider, among others, the investments and operations of energy technologies and the use of energy as well as the land-use change & management and yield increases. The individual choices of the mitigation portfolio share the characteristic that the emission abatement attributable to them is at or below a common marginal cost level. In an equilibrium context this solution could be implemented by an emission pricing system that covers the sectors, gases and regions as required in the optimal solution. As an alternative to carbon pricing the mitigation strategy could also be implemented by a set of comprehensive and well-balanced regulatory measures. Both the emission pricing and regulatory solution can be assumed to be equivalent in an optimal system where economic actors avoid actions leading to economic losses (optimality

condition). For the purpose of this study, we use the GHG price as an indicator for the level of policy ambition.

The models reach equilibrium through iterative runs of each individual model. After each run, REMIND reports GHG prices and bioenergy demand to MAgPIE. Subsequently, MAgPIE finds an optimal land-use pattern fulfilling the bioenergy demand and adds an incentive/disincentive to A/R / deforestation through the GHG price. MAgPIE, in turn, reports back emissions of the land sector and the price of delivering the required bioenergy, which is then accounted for in the next REMIND iteration (Supplementary Fig. 1).

MAgPIE is a partial equilibrium model minimizing the global cost of agricultural production. The demand for these products like food, feed, fibre, and timber depends on the socioeconomic development following SSP scenarios. Biophysical constraints for yield, water availability, and carbon density are provided by the vegetation model LPJmL (Supplementary Fig. 1). The expansion and reduction of forests in MAgPIE is driven by the demand for food, feed, fibre, and timber with cropland and pasture extensification leading to deforestation. Further, forest expansion and management are motivated by mitigation needs through the GHG price and by the demand for wood and woodfuel motivating plantation forests with dynamic rotation lengths. In addition, the GHG price also reduces deforestation by assigning a cost to the emissions generated from converting forests to other land uses. A 50-year planning horizon is used to establish the value of the mitigation forest. The removed carbon and the future carbon price are known in this decision. However, the revenue created by the future GHG price is discounted by the expected interest rate (Supplementary Data Table 1). Forests accumulate carbon along sigmoidal growth curves reaching their local storage potential provided by LPJmL.

LPJmL v4 is a process-based, dynamic global vegetation model simulating natural and agricultural ecosystem dynamics, including production, and stocks and fluxes of water and carbon[39]. It integrates gridded, daily climate model data including precipitation, temperatures, and radiation at a 0.5-degree resolution as well as global annual atmospheric $CO_2$ levels (Supplementary Fig. 1). The model represents both managed and unmanaged vegetation through 26 functional types of which ten represent tree types. Crop yields and irrigation water requirements as used by MAgPIE (Supplementary Fig. 1) are computed with a later model version (v5)[40] consistent with the data used by the Global Gridded Crop Model Intercomparison crop yield projections under climate change[41]. LPJmL has been continuously employed to study the effects of changing environmental conditions on both natural and managed ecosystems. It has undergone thorough validation against independent datasets for key variables like carbon stocks, fluxes, atmospheric $CO_2$ concentration, and vegetation cover[42] following established, standardized benchmarking systems described by Kelley et al. [43]. The validation protocol was further extended to include data from eddy flux towers, remote sensing, and FAOSTAT to assess the model performance regarding evapotranspiration rates, net ecosystem exchange and respiration rates, and crop yields[42]. Most pertinent for this study, terrestrial carbon dynamics are evaluated by eddy flux measurements for gross primary productivity while net primary production is evaluated by measurements of leaf-litter, stems, branches and roots. Vegetation carbon stocks are validated using microwave satellite and lidar observations and the model's fire dynamics are evaluated using satellite-based assessments of burnt area[42].

REMIND is a general equilibrium model combining a Ramsey-type growth model of the economy with an engineering-based energy system model. The two models link the final energy demand generated by the economic activity of major sectors like transport, industry, and buildings. In turn, the economic core is informed about the cost incurred by its energy use. The energy sector's transformation is constrained by inertia and path dependencies created by existing infrastructure and learning curves/adjustment costs incurred by

adopting new technologies. Emissions of major GHGs are linked to the primary energy sources.

The simple climate model MAGICC v7.5.3[44] is used to estimate a range of climate and carbon cycle outcomes of the derived emissions scenarios. MAGICC includes simple representations of atmosphere, land and ocean processes. The model is run 600 times for each emissions scenario with different parametrizations, calibrated and constrained to emulate the probabilistic behavior of the ensemble of Earth System Models (ESMs) used in several model intercomparison projects part of CMIP6, including C4MIP and ScenarioMIP[45]. The harmonization and infilling procedures, as well as the parametrization used here are identical to those used in the climate assessment of the IPCC Sixth assessment report of mitigation pathways[46] (see Kikstra et al. 2022).

Peak cumulated emission budgets of 500GtCO$_2$ and 1150GtCO$_2$ aim at peak warmings of 1.5 °C and 2 °C. The temperature is allowed to be at or slightly above its peak but is required to return below the warming threshold by 2100. More than 50% of 1.5 °C and 67% of 2 °C scenarios reach the threshold by the end of the century[47]. The scenarios establishing the two emission budgets are independent of the scenarios produced in this study. The difference between carbon released and regrown induced by the forest disturbance is accounted for under net land-use emissions taking up part of the emission budget.

## Modelled disturbances

LPJmL simulates I) Background mortality depending on the growth efficiency; II) Stress mortality created by competition from high tree density, and III) Mortality caused by crown and cambial damage of fires. The mortalities modelled in LPJmL determine the potential carbon density passed to the land-use allocation model MAgPIE which determines the forest carbon equilibrium and growth rate of MAgPIE. Thus, these mortalities do not lead to carbon emissions having only an indirect effect on the mitigation scenario of the full model chain (Supplementary Fig. 1) by lowering the potential carbon density.

MAgPIE does not simulate individual trees but represents forests as collections of different age classes that develop and accumulate carbon over time. Previously, forest carbon storage in MAgPIE only increased over time as the forest aged. In this study, a share of each age class is set back to the youngest class, releasing the carbon difference between them. This introduction of disturbances to MAgPIE's forest age classes enables the modeling of disturbance-driven emissions to directly impact mitigation scenarios.

Four disturbance rates are explored. Each higher level doubles the disturbance factor from two to four to eight to 16 per thousand per year. Each disturbance rate is introduced in 2030 and built up linearly until the full level is reached in 2050.

The disturbed share of the forest is allowed to regrow immediately. Thus, no forest area is lost to the disturbance. Primary forests are the exception to this as the land type is exclusive to pristine forests in the model. As such, disturbed primary forest is reclassified and regrown as secondary forest. Between perpetual disturbance and regrowth, forests will move towards a new equilibrium carbon storage.

REMIND-MAgPIE runs in timesteps of five years to 2060 and in ten-year timesteps onward to 2150. The yearly disturbance value is multiplied by the timestep length to form the disturbance's extent between two timesteps. The modified age-class distribution is allowed to age and accumulate carbon immediately. Thus, the released carbon from the loss of storage is partly offset by the regrowth produced by the first age class.

Because the disturbance is summarized over the whole timestep, all yearly disturbances within that time are allowed to regrow the full length of the timestep. Hence, the resulting carbon release is slightly smaller than an implementation where disturbances could be introduced yearly.

## Observed disturbances

Future disturbance levels are still highly uncertain, especially given the wide variety of biotic and abiotic sources modified by climate change accompanying degradation and clearing. Nevertheless, historic disturbance rates and their progression can provide insight, albeit limited to current climatic conditions. Historic disturbance rates in Europe and Canada are estimated between 0.26% per yr and 0.57% per yr[20,48–50]. However, only the highest (0.57% per yr) assessment included direct human interventions[50]. Rates in the Amazon could already be much higher as untracked degradation alone was found to surpass even deforestation by now affecting an average of 0.37% of forest area per year over the 22-year study period[51]. Finding a marked increase in historic disturbance rates over the last decades[20,21,48,50,52–55], a future increase under accelerating climate change and resource demand-induced pressure is deemed likely[16,22,48,52,56–59]. Increases in disturbance rates were found globally except for the Congo basin, where no significant increase was observed[52]. Moreover, recently observed spikes in single-agent disturbances could indicate non-linear behaviour in response to direct and indirect human-induced stress[20–22,54,57]. Bark beetle disturbance rates doubled in Europe over the last 20 years[20]. Further, bark beetle damage in the Czech Republic surged from a long-term range of 0.2% per yr - 1.4% per yr to up to 5.4% per yr in most recent years[21]. Similarly, disturbance related to declining forest health overtook all other types of disturbances in the US since the late 1990s, increasing dramatically from 0.07% per yr in 1985 to 2.82% per yr in 2001[54].

## Scenario setup

Five policy response scenarios investigate each disturbance rate, differing in their response to the impermanent carbon storage in forests. The scenarios are shaped to explore the space of possible reactions to forest impermanence ranging from immediate, proactive actions to myopic scenarios in which action is only taken years after the forest disturbance is evident.

The most optimistic scenario assumes perfect foresight of the disturbance. Therefore, actions are taken immediately, preparing for forest impermanence even before they take effect in 2030.

Two less optimistic scenarios do not prepare in advance and only act after the forest disturbance becomes evident in 2030. The first/second scenario acts in the first/second timestep (2035/2040) after carbon is first released from forests delaying action by five/ten years. However, after the initial delay, the response is again drawn from a perfect knowledge of future disturbance rates.

The two least optimistic scenarios initially follow the five/ten-year delayed scenarios. However, instead of the perfect foresight of increasing disturbance rates after the delay, they myopically only respond to the disturbance rate experience in that timestep. Thus, the rising disturbance rate between 2030 and 2050 is continuously underestimated. e.g., the myopic scenario acting with a five-year delay only responds in 2035 and as if the disturbance rate remains at its 2035 level of only 40% of its total value. Subsequently, with each timestep (five years), the action is adjusted to prepare as if the disturbance rate will stay at the newly reached point in time indefinitely.

Two cumulative emission peaks are investigated. The more stringent 500 GtCO$_2$ peak corresponds to the 1.5 °C target, and the 1150 GtCO$_2$ peak aims at the 2°C target detailed in the Paris Agreement.

All response scenarios are explored under middle-of-the-road socioeconomic conditions of the SSP2 pathway. They are compared to a reference SSP2 scenario without forest disturbances to investigate the necessary mitigation adjustments to deal with the disturbance and the cost this incurs. In addition, the response scenarios are also contrasted against each other to probe the cost and effort associated with neglecting disturbances.

To understand the uncertainty stemming from the underlying socioeconomic scenario, the policy responses explored in the main text are also assessed in the contexts of SSP1 and SSP5 as detailed in Supplementary Fig. 3.

## Data availability

The model output data generated in this study have been deposited in the Zenodo database under accession code [https://doi.org/10.5281/zenodo.14772622][60].

## Code availability

The source code for REMIND 3.0 is available at https://github.com/remindmodel/remind[61], together with the model documentation at https://rse.pik-potsdam.de/doc/remind/3.0.0/. The source code of MAgPIE 4.6 is available at https://github.com/magpiemodel[62], with the model documentation at https://rse.pik-potsdam.de/doc/magpie/4.6.0/.

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

## Acknowledgements

This project has received funding from the European Union's Horizon 2020 Research and Innovation Programme under Grant Agreement Nos 101003536 (ESM2025) and 101056939 (RESCUE).

## Author contributions

M.G.W., F.H., and A.P. conceived the study. M.G.W., F.H., L.M., N.B., and A.P. designed the scenarios. M.G.W. produced the REMIND-MAgPIE model runs with assistance from F.H. and L.M. MAgPIE code was extended by M.G.W. and F.H. M.G.W. drafted the manuscript and created the figures with significant input from F.H., L.M., N.B., G.L., J.P.D., J.H., G.A,. C.M., H.L.-C., and A.P.

## Funding

## Competing interests

The authors declare no competing interests.
