## [Transparent Peer Review file · Nature Communications]

Hedging our bet on forest permanence for the economic viability of climate targets

Corresponding Author: Dr Michael Windisch

Version 0:

Reviewer comments:

Reviewer #1

(Remarks to the Author)

The paper presents significant findings on the future projection of carbon stocks in forests under various forest disturbance and climate change (SSP) scenarios. It offers an integrated approach, including a model chain from a biophysical forest growth model to the MAgPIE land-use model. The study illustrates how CO₂ prices would vary under three different scenarios with stylized disturbance rates.

Although the approach towards future risks of natural disturbances is somewhat simplified, the results contribute to research on permanence. The work is original and provides insights into future scenarios for climate change mitigation. However, a major limitation is the simplified representation of the interaction between forests and natural disturbances, which are not modeled explicitly. As a result, the drivers of natural disturbances are assumed rather than modeled, and human interactions with forests and disturbances are not considered.

The work provides important conclusions, but the background is not fully realistic due to the stylized modeling of forest disturbances. For instance, the assumption of a linear increase in tree mortality from disturbances over the study period is unclear, as disturbance rates are typically time-dependent and interact with other socio-economic sectors. While wildfire models exist that can project forest fires under specific climate change scenarios, the authors only mention that these models fail to reproduce historical fires without providing a clear justification for their chosen scenarios. Nevertheless, the work supports its conclusions and claims.

The data analysis is well-documented; however, the manuscript would benefit from a more detailed description of how the forest model simulates natural disturbances and forest management. This additional detail would enhance the precision of the findings.

The methodology is sound but overly stylized. A more realistic representation of forest disturbances and their feedback on carbon stocks is necessary for a comprehensive analysis. It is also important to consider the impacts of forest management on disturbance rates and permanence, which appear to be overlooked in the biophysical model.

The modeling approach is complex, yet based on simplified assumptions. The concept of CO₂ pricing should be clarified further. Specifically, what is the CO₂ price, how is it defined in the model, what is the current CO₂ price, and is it accepted by carbon markets?

The paper has the potential to make a significant contribution to the field, but the authors should better emphasize the limitations of their modeling approach and provide a clearer definition of the CO₂ price.

(Remarks on code availability)

Reviewer #2

(Remarks to the Author)

The manuscript "Hedging our bet on forest permanence" presents interesting perspectives on the role of forest carbon sinks in realising the goals of the Paris Agreement.

The authors have dealt with a very important topic of non-permanence of forest carbon stocks in the light of a changing climate leading to increased pests, diseases and fire. While this is an important contribution, it would be of value to have scenarios that also consider differential rates of pests, diseases and fire in a changing climate to highlight the challenges and the limitations of global carbon budget estimates. This in my opinion would highlight the urgency for formulating and implementing policy measures, and would also signal the need for more rigorous and aggressive measures to achieve the Paris Agreement goals and targets.

If this is beyond the scope of the modelling framework adopted, some additional discussion highlighting the need for such work would add value to the paper.

decision: Accept with minor changes.

(Remarks on code availability)

Reviewer #3

(Remarks to the Author)

Summary

Keeping to the goal of 1.5 degrees Celsius warming, Windisch et al. model the implications of a range of more realistic direct and indirect human disturbance rates on forest land as well as several climate projections, socioeconomic possibilities, and policy responses on all sectors. Results are framed in terms of if the response is more immediate or more delayed and suggest that in the face of uncertain forest carbon stock and sequestration emission reductions in other sectors must be immediate and ambitious to avoid greater cost (i.e., carbon prices and GDP loss) and effort (i.e., more mitigation land). The study is timely, as forests are increasingly relied on in climate mitigation strategies across jurisdictional scales. A couple of general comments before line by line comments.

There are figures in the supplementary information that include uncertainty or are results of a sort of sensitivity analysis. It would benefit the main text if some of this information could be incorporated into the main text, perhaps Fig 1a or 5.

The authors should consider including more background on current methods, what models are used, what is included and what is not, being explicit the scale (i.e., global) and that different scales (e.g., national) use different methods and definitions, e.g., the managed land proxy, in terms of accounting (e.g., the Global Stocktake and National GHG Inventories) and anticipation (e.g., Nationally Determined Contributions) of forest's contribution to climate mitigation. A couple suggested (not required) references to get started:

Grassi, G., Schwingshackl, C., Gasser, T., Houghton, R. A., Sitch, S., Canadell, J. G., Cescatti, A., Ciais, P., Federici, S., Friedlingstein, P., Kurz, W. A., Sanz Sanchez, M. J., Abad Viñas, R., Alkama, R., Bultan, S., Ceccherini, G., Falk, S., Kato, E., Kennedy, D., Knauer, J., Korosuo, A., Melo, J., McGrath, M. J., Nabel, J. E. M. S., Poulter, B., Romanovskaya, A. A., Rossi, S., Tian, H., Walker, A. P., Yuan, W., Yue, X., and Pongratz, J.: Harmonising the land-use flux estimates of global models and national inventories for 2000–2020, *Earth Syst. Sci. Data*, 15, 1093–1114, <https://doi.org/10.5194/essd-15-1093-2023>, 2023.

Nabuurs, G.J., Ciais, P., Grassi, G. et al. Reporting carbon fluxes from unmanaged forest. *Commun Earth Environ* 4, 337 (2023). <https://doi.org/10.1038/s43247-023-01005-y>

Comment line by line

12: Instead of 'climate objectives' consider emission reduction goals.

12-13: Instead of 'expanding forests' consider enhancing the forest carbon sink. The way it is currently phrase suggests it's an issue of land use/land cover, which might only be part of the solution.

31: Consider incorporating the most recent analysis: Pan, Y., Birdsey, R.A., Phillips, O.L. et al. The enduring world forest carbon sink. *Nature* 631, 563–569 (2024). <https://doi.org/10.1038/s41586-024-07602-x>

32: Need more background on the managed land proxy here. What assumptions are made? What processes (natural vs anthropogenic) are realistically included? Consider the work of Giacomo Grassi, for example:

53-54: Consider including references on the assumptions of climate envelope models and whether the framework of the leading vs the trailing edge is evidence-based.

55-56: What disturbance source does phosphorus limitation fit under? An indirect human disturbance? I'd suggest removing this example. It's confusing. Unless the authors explain the connection to human intervention.

59: Is there a specific number missing in front of years?

61: Be explicit about the disturbance sources: natural disturbance (insects and disease), indirect human disturbance (climate change, CO₂), direct human disturbance (harvest). Could also mention the interaction between these sources of disturbance is complex to model and sometimes unknown.

68: Missing in projections at the global scale or in the global carbon budget. Direct and indirect human disturbances aren't missing at the national scale.

148: Consider faceting like Fig 4a. It would be more intuitive and easier for readers to compare the differences if all scenarios were on one figure. If the figure is not faceted, add more explanation in the caption on if difference is negative/positive what it means.

111. No Fig 6.

201: No Fig. 6.

206-207: The way the sentence is structured, it reads that GCP cost doubles

208: No Fig. 6.

210: Is there supposed to be a dotted line at 2100 in Fig 5?

230-231: Add citation.

(Remarks on code availability)

Version 1:

Reviewer comments:

Reviewer #2

(Remarks to the Author)

Concerns raised by me during the first review are well addressed.

(Remarks on code availability)

Comments have been addressed adequately, I recommend publication.

Reviewer #3

(Remarks to the Author)

I appreciate the significant effort by the authors to meaningfully address all reviewer suggestions and comments. All my concerns were addressed and I have no suggestions or comments to consider.

(Remarks on code availability)

We truly appreciate the time and effort each reviewer invested into reviewing our manuscript. The constructive nature of the comments and insights were very helpful and, we believe, have allowed us to strengthen the manuscript.

Below, we've provided a point-by-point response (blue) to each comment (black), outlining the revisions made and explaining the rationale behind them.

REVIEWER COMMENTS

Reviewer #1 (Remarks to the Author):

The paper presents significant findings on the future projection of carbon stocks in forests under various forest disturbance and climate change (SSP) scenarios. It offers an integrated approach, including a model chain from a biophysical forest growth model to the MAgPIE land-use model. The study illustrates how CO₂ prices would vary under three different scenarios with stylized disturbance rates.

Although the approach towards future risks of natural disturbances is somewhat simplified, the results contribute to research on permanence. The work is original and provides insights into future scenarios for climate change mitigation. **However, a major limitation is the simplified representation of the interaction between forests and natural disturbances, which are not modeled explicitly.** As a result, the drivers of natural disturbances are assumed rather than modeled, and **human interactions with forests and disturbances are not considered.**

The work provides important conclusions, but the background is not fully realistic due to the stylized modeling of forest disturbances. For instance, the assumption of a linear increase in tree mortality from disturbances over the study period is unclear, as disturbance rates are typically time-dependent and interact with other socio-economic sectors. While wildfire models exist that can project forest fires under specific climate change scenarios, the authors only mention that these models fail to reproduce historical fires without providing a clear justification for their chosen scenarios. Nevertheless, the work supports its conclusions and claims.

The data analysis is well-documented; however, the manuscript would benefit from a more detailed description of how the forest model simulates natural disturbances and forest management. This additional detail would enhance the precision of the findings.

We've expanded the method section on both the modeling of timber products (wood, woodfuel) and the modeling of disturbances both in the vegetation model LPJmL that delivers the potential carbon densities and the explanation of disturbances in MAgPIE specifically highlighting the novelty and use of the chosen approach. The two new sections state the following:

Lines 443 – 448:

“The expansion and reduction of forests in MAgPIE is driven by the demand for food, feed, fibre, and timber with cropland and pasture extensification leading to

deforestation. Further, forest expansion and management are motivated by mitigation needs through the GHG price and by the demand for wood and woodfuel motivating plantation forests with dynamic rotation lengths. In addition, the GHG price also reduces deforestation by assigning a cost to the emissions generated from converting forests to other land uses.”

Lines 510 – 524:

“LPJmL simulates I) Background mortality depending on the growth efficiency; II) Stress mortality created by competition from high tree density, and III) Mortality caused by crown and cambial damage of fires. The mortalities modelled in LPJmL determine the potential carbon density passed to the land-use allocation model MAgPIE which determines the forest carbon equilibrium and growth rate of MAgPIE. Thus, these mortalities do not lead to carbon emissions having only an indirect effect on the mitigation scenario of the full model chain (Fig. S1) by lowering the potential carbon density.

MAgPIE does not simulate individual trees but represents forests as collections of different age classes that develop and accumulate carbon over time. Previously, forest carbon storage in MAgPIE only increased over time as the forest aged. In this study, a share of each age class is set back to the youngest class, releasing the carbon difference between them. For the first time, this introduction of disturbances to MAgPIE’s forest age classes enables the modeling of disturbance-driven emissions to directly impact mitigation scenarios.”

The methodology is sound but overly stylized. A more realistic representation of forest disturbances and their feedback on carbon stocks is necessary for a comprehensive analysis. It is also important to consider the impacts of forest management on disturbance rates and permanence, which appear to be overlooked in the biophysical model.

We agree that our approach does not deliver an improved forest disturbance projection. Dedicated, process-based models are needed for this task. Neither can it deliver on potential improved forest management techniques to reduce said disturbances. However, the introduced, stylized disturbances here are, to our knowledge, the first time disturbances and the effects of potential forest impermanence on mitigation scenarios are assessed by a global, multi-sectoral model. Thus, they are the first to bring forward the importance of forest permanence considerations especially to low warming mitigation pathways.

To make this novelty but also its limitations clearer we have taken the following five steps:

1. To provide context for this type of global, multi-sectoral modelling within the climate discussion, we have added a paragraph to the discussion section.

Lines 338 – 350:

“Discussed in this study are aspects and shortcomings of global models that estimate current and future carbon emissions and removals from forests informing and simulating mitigation pathways. The carbon flux estimates of these global models are distinctly different from the country-level Nationally Determined Contributions (NDCs) and their tracking by the National Greenhouse Gas Inventories (NGHGs) as their definition of anthropogenic vs natural forest carbon fluxes differs^{34–36}. Global mitigation pathway simulations highlighted in the IPCC reports are produced by IAMs, including the REMIND-MAGPIE model employed in this study. Most commonly, these mitigation pathways of IAMs only consider emissions and removals induced by direct anthropogenic effects such as deforestation, wood harvest, and the regrowth that follows harvest or land abandonment. In contrast, the NGHGs tasked to track the NDCs consider all carbon fluxes in “managed forests”, which serves as a proxy to distinguish between anthropogenic and natural carbon emissions and removals. Thus, the highly stylized disturbance regimes used in this study provide an initial assessment of the role forest disturbances play within the global multi-sectoral mitigation modeling of IAMs.”

2. To improve robustness of the conveyed results, we now emphasize findings that are consistent across all assessed disturbance rates in all key sections of the manuscript (abstract, introduction, results, discussion). These findings result from interactions between the major emission and mitigation action simulated by the model and could not have been deduced without the global, multi-sectoral modeling approach of this study.

Lines 17-21; Lines 93 -116; Lines 130-134; Lines 179-186; Lines 222-227; Lines 284-291; Lines 304-310; Lines 332-336.

3. To be more upfront with the limitations of this stylized approach, we’ve added a paragraph to the discussion section stating that our approach can only provide an initial assessment and the need for further developments of future disturbance projections to achieve a more realistic outlook.

Lines 348 – 355:

“Thus, the highly stylized disturbance regimes used in this study provide an initial assessment of the role forest disturbances play within the global multi-sectoral mitigation modeling of IAMs. A more detailed understanding of both direct and indirect human influences on these disturbances is essential for incorporating more realistic representations into upcoming multi-sectoral modeling efforts. Additionally, examining different rates of forest disturbances caused by individual drivers, such as pests or windfall—for example, assessing a strong pest versus a strong windthrow response to climate change—could provide more targeted insights to inform management strategies for mitigating specific types of disturbances.”

4. To highlight that the stylized disturbance rates are within the range of observed disturbance rates, we’ve added a new figure (Fig. S7) and table alongside the

method section on “Observed disturbances” (line 545 ff.) that visually show various disturbance estimates in comparison to the disturbance level of this study.

5. In response to the previous comment, we’ve elaborated in two additional sections on the forest dynamics in MAgPIE and the (previous) representation of disturbances and permanence in LPJmL and MAgPIE (Lines 443 – 448 & Lines 510 – 524).

The modeling approach is complex, yet based on simplified assumptions. The concept of CO2 pricing should be clarified further. Specifically, what is the CO2 price, how is it defined in the model, what is the current CO2 price, and is it accepted by carbon markets?

The paper has the potential to make a significant contribution to the field, but the authors should better emphasize the limitations of their modeling approach and provide a clearer definition of the CO2 price.

We’ve added a paragraph to the method section detailing the theoretical background of deriving climate strategies in the model including the meaning of the CO2 price and its specific use and applicability as an indicator in this study. Additionally, we’ve expanded on how the CO2 price applies specifically to the land-use and forestry sector.

Lines 419 – 432:

“The REMIND and the MAgPIE models compute economic equilibrium by combining optimization methods and iterative fixed-point methods. For a given climate target, such as a carbon budget that constrains global cumulative emissions, the coupled model framework derives the optimal strategy given the socioeconomic, technological, political and geobiophysical constraints. This optimal strategy is a portfolio of mitigation measures that consider, among others, the investments and operations of energy technologies and the use of energy as well as the land-use change & management and yield increases. The individual choices of the mitigation portfolio share the characteristic that the emission abatement attributable to them is at or below a common marginal cost level. In an equilibrium context this solution could be implemented by an emission pricing system that covers the sectors, gases and regions as required in the optimal solution. As an alternative to carbon pricing the mitigation strategy could also be implemented by a set of comprehensive and well-balanced regulatory measures. Both the emission pricing and regulatory solution can be assumed to be equivalent in an optimal system where economic actors avoid actions leading to economic losses (optimality condition). For the purpose of this study, we use the GHG price as an indicator for the level of policy ambition.”

&

Lines 443 – 451:

“The expansion and reduction of forests in MAGPIE is driven by the demand for food, feed, fibre, and timber with cropland and pasture extensification leading to deforestation. Further, forest expansion and management are motivated by mitigation needs through the GHG price and by the demand for wood and woodfuel motivating plantation forests with dynamic rotation lengths. In addition, the GHG price also reduces deforestation by assigning a cost to the emissions generated from converting forests to other land uses. A 50-year planning horizon is used to establish the value of the mitigation forest. The removed carbon and the future carbon price are known in this decision. However, the revenue created by the future GHG price is discounted by the expected interest rate (Supplementary Data Table 1).”

Further, also in response to the last comment and comments by reviewer 3, we’ve added additional context to how this type of global modeling fits into the ongoing discussion of forest carbon fluxes (please kindly refer to our response to the previous comment, subpoint 1., lines 338 – 350)

In addition, we have added a paragraph that state the limitations and potential future improvements (kindly refer to our response to the previous comment, subpoint 3., Lines 348 – 355).

Reviewer #2 (Remarks to the Author):

The manuscript "Hedging our bet on forest permanence" presents interesting perspectives on the role of forest carbon sinks in realising the goals of the Paris Agreement.

The authors have dealt with a very important topic of non-permanence of forest carbon stocks in the light of a changing climate leading to increased pests, diseases and fire. While this is an important contribution, it would be of value to have scenarios that also consider differential rates of pests, diseases and fire in a changing climate to highlight the challenges and the limitations of global carbon budget estimates. This in my opinion would highlight the urgency for formulating and implementing policy measures, and would also signal the need for more rigorous and aggressive measures to achieve the Paris Agreement goals and targets.

If this is beyond the scope of the modelling framework adopted, some additional discussion highlighting the need for such work would add value to the paper.

decision: Accept with minor changes.

We thank the reviewer for acknowledging the contribution of this manuscript. Also in response to comments of reviewer 1, we now emphasize findings that remain consistent across the full range of assessed disturbance rates in all key sections of the manuscript (abstract, introduction, results, discussion). We agree that this can further

highlight the urgency of monitoring and safeguarding forests to still achieve the Paris Agreement's goals.

In the following, we highlight two key sentences added to the manuscript. Noting all changed sections with line numbers below:

Lines 93 – 116:

“All additional disturbance rates introduced lie within the range of observed rates (see Methods & Fig. S7). Moreover, we emphasize findings that remain consistent across the entire range of disturbance rates.”

Lines 17 – 21:

“However, delaying action on forest carbon loss by just five years consistently doubles the additional mitigation costs and efforts across key sectors, regardless of the assessed forest disturbance rates. Moreover, these myopic responses to forest carbon loss are as stringent, or even more demanding, than immediate responses to twice the forest disturbance rate.”

& Lines 130-134; Lines 179-186; Lines 222-227; Lines 284-291; Lines 304-310; Lines 332-336.

Further, we now state in the discussion section the limitations of our stylized approach and potential future improvements, especially regarding different rates of disturbance drivers, in the following:

Lines 348 – 355:

“Thus, the highly stylized disturbance regimes used in this study provide an initial assessment of the role forest disturbances play within the global multi-sectoral mitigation modeling of IAMs. A more detailed understanding of both direct and indirect human influences on these disturbances is essential for incorporating more realistic representations into upcoming multi-sectoral modeling efforts. Additionally, examining different rates of forest disturbances caused by individual drivers, such as pests or windfall—for example, assessing a strong pest versus a strong windthrow response to climate change—could provide more targeted insights to inform management strategies for mitigating specific types of disturbances.”

Reviewer #3 (Remarks to the Author):

Summary

Keeping to the goal of 1.5 degrees Celsius warming, Windisch et al. model the implications of a range of more realistic direct and indirect human disturbance rates on forest land as well as several climate projections, socioeconomic possibilities, and policy responses on all sectors. Results are framed in terms of if the response is more

immediate or more delayed and suggest that in the face of uncertain forest carbon stock and sequestration emission reductions in other sectors must be immediate and ambitious to avoid greater cost (i.e., carbon prices and GDP loss) and effort (i.e., more mitigation land). The study is timely, as forests are increasingly relied on in climate mitigation strategies across jurisdictional scales. A couple of general comments before line by line comments.

There are figures in the supplementary information that include uncertainty or are results of a sort of sensitivity analysis. It would benefit the main text if some of this information could be incorporated into the main text, perhaps Fig 1a or 5.

Also in response to comments of reviewers 1 and 2, we now incorporate findings from the sensitivity analysis of Fig. S2 & S3 in all key sections of the manuscript (abstract, introduction, results, discussion). These results remain consistent across the full range of assessed disturbance rates.

In the following, we highlight two key sentences added to the manuscript. Noting all changed sections with line numbers below:

Lines 93 – 116:

“All additional disturbance rates introduced lie within the range of observed rates (see Methods & Fig. S7). Moreover, we emphasize findings that remain consistent across the entire range of disturbance rates.”

Lines 17 – 21:

“However, delaying action on forest carbon loss by just five years consistently doubles the additional mitigation costs and efforts across key sectors, regardless of the assessed forest disturbance rates. Moreover, these myopic responses to forest carbon loss are as stringent, or even more demanding, than immediate responses to twice the forest disturbance rate.”

& Lines 130-134; Lines 179-186; Lines 222-227; Lines 284-291; Lines 304-310; Lines 332-336.

The authors should consider including more background on current methods, what models are used, what is included and what is not, being explicit the scale (i.e., global) and that different scales (e.g., national) use different methods and definitions, e.g., the managed land proxy, in terms of accounting (e.g., the Global Stocktake and National GHG Inventories) and anticipation (e.g., Nationally Determined Contributions) of forest's contribution to climate mitigation. A couple suggested (not required) references to get started:

Grassi, G., Schwingshackl, C., Gasser, T., Houghton, R. A., Sitch, S., Canadell, J. G., Cescatti, A., Ciais, P., Federici, S., Friedlingstein, P., Kurz, W. A., Sanz Sanchez, M. J., Abad Viñas, R., Alkama, R., Bultan, S., Ceccherini, G., Falk, S., Kato, E., Kennedy, D., Knauer, J., Korosuo, A., Melo, J., McGrath, M. J., Nabel, J. E. M. S., Poulter, B., Romanovskaya, A. A., Rossi, S., Tian, H., Walker, A. P., Yuan, W., Yue, X., and Pongratz, J.: Harmonising the land-use flux estimates of global models and national inventories

for 2000–2020, *Earth Syst. Sci. Data*, 15, 1093–1114, <https://doi.org/10.5194/essd-15-1093-2023>, 2023.

Nabuurs, G.J., Ciais, P., Grassi, G. et al. Reporting carbon fluxes from unmanaged forest. *Commun Earth Environ* 4, 337 (2023). <https://doi.org/10.1038/s43247-023-01005-y>

We agree with the reviewer that these important and heavily discussed issues are an important addition that help put into context this specific manuscript against the backdrop of discussion around forest carbon fluxes in global modelling and the sphere of the NGHGs, Global Stocktake, and NDCs.

We have added a paragraph detailing both distinct approaches of estimating anthropogenic forest carbon fluxes to the discussion section.

Lines 338 – 350:

“Discussed in this study are aspects and shortcomings of global models that estimate current and future carbon emissions and removals from forests informing and simulating mitigation pathways. The carbon flux estimates of these global models are distinctly different from the country-level Nationally Determined Contributions (NDCs) and their tracking by the National Greenhouse Gas Inventories (NGHGs) as their definition of anthropogenic vs natural forest carbon fluxes differs^{34–36}. Global mitigation pathway simulations highlighted in the IPCC reports are produced by IAMs, including the REMIND-MAGPIE model employed in this study. Most commonly, these mitigation pathways of IAMs only consider emissions and removals induced by direct anthropogenic effects such as deforestation, wood harvest, and the regrowth that follows harvest or land abandonment. In contrast, the NGHGs tasked to track the NDCs consider all carbon fluxes in “managed forests”, which serves as a proxy to distinguish between anthropogenic and natural carbon emissions and removals. Thus, the highly stylized disturbance regimes used in this study provide an initial assessment of the role forest disturbances play within the global multi-sectoral mitigation modeling of IAMs.”

Comment line by line

12: Instead of ‘climate objectives’ consider emission reduction goals.

Thank you for this suggestion. We agree that the proposed wording is more precise than our initial version and have updated it accordingly (line 12).

12-13: Instead of ‘expanding forests’ consider enhancing the forest carbon sink. The way it is currently phrase suggests it’s an issue of land use/land cover, which might only be part of the solution.

We appreciate the input; Again, the suggested phrasing better captures our intent. We’ve modified it to “enhancing carbon storage and sequestration” to be explicit about the scope of the study being about both the existing forest carbon storage and plans to enhance it (lines 12-13).

31: Consider incorporating the most recent analysis: Pan, Y., Birdsey, R.A., Phillips, O.L.

et al. The enduring world forest carbon sink. Nature 631, 563–569 (2024). <https://doi.org/10.1038/s41586-024-07602-x>

Thank you for spotting that one of the initial references has now an updated estimate for the global forest carbon sink. We've changed the estimate and reference (line 50).

32: Need more background on the managed land proxy here. What assumptions are made? What processes (natural vs anthropogenic) are realistically included? Consider the work of Giacomo Grassi, for example:

Thank you for pointing us to closer examine these statements again. Considering the terminology surrounding the discussion between anthropogenic and natural carbon fluxes in national inventories vs. global models, we realized that these sentences were ambiguous at best. We have removed the statements on what was initially called “unassisted” or “natural” sinks and have added the previously mentioned paragraph on anthropogenic vs. natural forest fluxes and how they are estimated in national inventories versus global models (lines 338 – 350).

53-54: Consider including references on the assumptions of climate envelope models and whether the framework of the leading vs the trailing edge is evidence-based.

We now include the following sentence based on the work of Warszawski et al. (“A multi-model analysis of risk of ecosystem shifts under climate change”, 10.1088/1748-9326/8/4/044018)

Lines 70 – 72:

“Global vegetation models agree that severe ecosystem changes, including shifting ecosystem boundaries, will increase with higher warming levels; however, the specific regions at risk remain uncertain.”

55-56: What disturbance source does phosphorus limitation fit under? An indirect human disturbance? I'd suggest removing this example. It's confusing. Unless the authors explain the connection to human intervention.

We have added a clarification that phosphorous limitation is found to limit the CO₂ fertilization effect in the Amazon. Thus, limiting an indirect human influence enhancing the forest carbon sink that models currently neglect.

Lines 74 – 77:

The statement now reads: “Further, models project the Amazon rainforest to act as a major carbon sink that may even increase in the future due to higher atmospheric CO₂ concentrations. However, more recent studies provide direct evidence of phosphorous limitation potentially cutting biomass carbon growth from CO₂ fertilization in half.”

59: Is there a specific number missing in front of years?

We've now changed the phrase to "a few years" matching the term in the cited study of Pendrill et al. 2022 (line 80).

61: Be explicit about the disturbance sources: natural disturbance (insects and disease), indirect human disturbance (climate change, CO₂), direct human disturbance (harvest). Could also mention the interaction between these sources of disturbance is complex to model and sometimes unknown.

We agree that this is a valuable clarification and, thus, have specified the disturbance source not only in line 61 but the throughout the entire paragraph (lines 65 - 89). Additionally, we've stressed again the context of global modeling in response to the reviewer's earlier comments.

The section most changed by this now reads (lines 81 - 86):

"IAMs tasked with simulating mitigation pathways may have overestimated the remaining carbon budgets by neglecting these major disturbances. These include natural disturbances (windfall, pests, diseases, etc.), their modulation by indirect human disturbances, such as a warming climate and elevated CO₂ levels, and the underestimated influence of direct human disturbances from land expansion and harvest. This potential overestimation of the future forest carbon sink in IAMs suggests the need for precautionary adjustments to modeled mitigation targets."

68: Missing in projections at the global scale or in the global carbon budget. Direct and indirect human disturbances aren't missing at the national scale.

Thank you for this clarification. We now specifically state that these disturbances are missing in "...projections of IAMs" (line 93).

148: Consider faceting like Fig 4a. It would be more intuitive and easier for readers to compare the differences if all scenarios were on one figure. If the figure is not faceted, add more explanation in the caption on if difference is negative/positive what it means.

We've considered the suggested faceting to include the "control" scenario without FCL similar to Fig. 4a. However, the land-use change shown in Fig. 4a happens to be of a similar size both over time and between scenarios compared to the total value. This is different in the case of Fig. 3 showing supplied energy and emissions where the differences between scenarios happen to be much smaller than the total values. Thus, in Fig. 3 it would be difficult to see the difference between scenarios if we applied the same figure structure.

We've added two clarifying statements including a word of caution for the different y-axes:

Lines 235 – 236:

"Positive/negative values in (b) indicate more/less energy supplied compared to the SSP2-1.5°C scenario shown in (a) without FCL."

Lines 240 – 242:

“The negative values in (d) indicate reduced sectorial emissions compared to the SSP2-1.5°C scenario shown in (c) without FCL.”

“Please note the differences in the y-axis scales between (a) and (b), & (c) and (d).”

111. No Fig 6.

We are sorry for this oversight. Figure 4 a) and b) were two figures initially. We’ve updated the wrong figure numbers.

201: No Fig. 6.

Please see the first response to missing figure 6.

206-207: The way the sentence is structured, it reads that GCP cost doubles

We’ve rephrase this statement to hopefully make clearer that the additional increase, meaning the increment in response to FCL, doubles.

Lines 302 – 304:

“Thus, the additional increase in carbon price and GDP cost incurred by the FCL response more than doubles between the foresighted and myopic scenarios facing the same forest disturbance rate (Figs. 2, 5, & S6).”

208: No Fig. 6.

Please see the first response to missing figure 6.

210: Is there supposed to be a dotted line at 2100 in Fig 5?

We’ve added the missing dotted line (line 311).

230-231: Add citation.

We now cite Strassburg et al. 2018 (<https://doi.org/10.1038/s41559-018-0743-8>) on the interaction of biodiversity and forestation (line 363).